# Characterization of the Use of Emergency Contraception from Sentinel Pharmacies in a Region of Southern Europe

**DOI:** 10.3390/jcm10132793

**Published:** 2021-06-25

**Authors:** Anna M. Jambrina, Pilar Rius, Pilar Gascón, Mercè Armelles, Mariona Camps-Bossacoma, Àngels Franch, Manel Rabanal

**Affiliations:** 1Directorate-General for Healthcare Planning and Regulation, Ministry of Health, Government of Catalonia, 08028 Barcelona, Spain; amjambrina@gencat.cat (A.M.J.); m.armelles@gencat.cat (M.A.); 2Physiology Section, Department of Biochemistry and Physiology, Faculty of Pharmacy and Food Science, University of Barcelona, 08028 Barcelona, Spain; marionacampsb@gmail.com (M.C.-B.); angelsfranch@ub.edu (À.F.); 3Council of the Pharmacist’s Association of Catalonia, 08009 Barcelona, Spain; prius@ccfc.cat; 4Blanquerna School of Health Sciences, Ramon Llull University, 08022 Barcelona, Spain; mariapilargl@blanquerna.url.edu

**Keywords:** emergency contraception, contraceptive method, levonorgestrel, ulipristal acetate, community pharmacies, health services administration

## Abstract

Numerous studies have been published suggesting that emergency contraception (EC) is used repeatedly, but a lack of information regarding the profile of users makes it difficult to evaluate actual consumer habits. The aim of this study was to obtain information regarding the profile of users who obtain EC and other factors that might play a role, and to provide criteria to evaluate and improve the strategies of current contraceptive programs. This was an observational one-year study based on surveillance data on the provision of EC to women of reproductive age in 60 community pharmacies in Catalonia, Spain. In total, 941 notifications of dispensation of EC in Catalonia were received. A total of 44.2% of users said it was not the first time that they had taken the medication (repeat user). The percentage of users who used condoms was lower in repeat users compared to first-time users (56.7% vs. 64.4%, *p* < 0.05). A total of 25.7% of users stated that they did not use any barrier contraceptive method. The use of natural methods in repeat users was 53.8% in the subgroup who requested the medication after 48 h, significantly higher than in users who obtained the medication within the first 24 h (*p* < 0.05). A high percentage of repeat users with risky sexual behaviors were detected, suggesting that new measures must be implemented to provide information for this method, together with educational and preventive strategies.

## 1. Introduction

The primary aim of emergency contraception (EC) is to prevent an unwanted pregnancy following unprotected sexual intercourse. Its use has no medical contraindications and it is indicated mainly in women of reproductive age who had unprotected sex, either because they did not use contraception or the contraception failed due to inadequate or incorrect use, or because of sexual assault in which the woman was not protected by an effective contraceptive method [1,2,3,4].

The criteria for prescribing this medication are that it must not be used routinely and must never be considered a regular method of contraception, since users are not protected against sexually-transmitted diseases (STDs), including the human immunodeficiency virus (HIV) [1,2,3,4].

The medicinal products currently used in Spain as EC are post-coital pills, which contain levonorgestrel (LNG) and ulipristal acetate (UPA) as the active substances. They obtained marketing authorization in 2001 and 2009, respectively, and were initially subject to medical prescription. It was not until the end of 2009, in the case of LNG, and 2015, in the case of UPA, that they were dispensed over the counter. This allowed the population to have increased access to the medication and accorded the community pharmacist a more active role and greater responsibility in the indication and dispensing of EC [4].

For this reason, numerous practical guidelines on EC have been published in Spain since 2009, in order to ensure the rational use of the medication, guide and standardize the actions of community pharmacists to meet the demand for provision, and improve educational and preventive aspects so that users adopt safe sexual behavior [4,5,6].

In Europe, the World Health Organization (WHO) recommends the administration of EC as a single-dose pill that contains 1.5 mg of LNG or 30 mg of UPA. Both drugs affect the ovulation process, delaying follicular rupture, with no effects with respect to implantation of the ovum. This indicates that the efficacy of the medication decreases as time passes after unprotected intercourse. It has also been shown that, if administered after implantation of the ovum, EC is not effective at the doses indicated [1,2,3,4].

With regard to the safety of this medication, the side effects described in the product information sheets are mainly mild and transient. The most common adverse reactions are the onset of nausea and changes in the time and type of bleeding in the following menstrual period. Common adverse reactions have been reported, such as headache, myalgia, and fatigue, which typically disappear 48 h after administration of the medication. Uncommon adverse reactions identified include anorexia, ectopic pregnancy, exanthema, miscarriage, and weight gain. Multiple serious adverse events, including convulsion, ectopic pregnancy, febrile neutropenia, stroke, abdominal hernia, anaphylaxis, cancer, ovarian cyst rupture, serious infections, and suicidal ideation, have been reported [7,8].

Numerous studies have been published suggesting that this contraceptive method is used repeatedly, but a lack of information regarding the profile of users and their perception of EC as an urgent requirement makes it difficult to evaluate actual consumer habits [9]. For this reason, data that validate these suspicions need to be collected.

Currently, there are new forms of epidemiological surveillance in our setting that seek to improve the effectiveness and efficacy of detection systems and to incorporate other health indicators, such as population behavior and habits. Information on processes linked to these indicators is usually sparse or incomplete, and is typically difficult to identify with traditional surveillance tools. These processes require the integration of several agents with different professional profiles as well as having detection tools closer to the site of demand or where the event occurs. In this sense, the community pharmacist has enormous potential as a first-line agent to report the rational use of medicinal products and to educate the population to adopt safe sexual behavior [10].

In Catalonia, a region of Spain situated in South Europe, a pilot project on sentinel pharmacies, carried out in 2016, studied the dispensation of EC. The information recorded obtained the initial data on the profile of users and their patterns of behavior [11]. Nevertheless, a larger study is needed to obtain data that will allow us to develop new health indicators generated by the care dynamics of the pharmacy, reinforce traditional surveillance systems, and provide an overview of the use of this method.

Thus, the aim of this study was to obtain information regarding the profile of users who obtain EC and other factors that may be involved, and to provide criteria to evaluate and improve the strategies of current contraceptive programs.

## 2. Materials and Methods

This was a descriptive, observational, prospective, one-year study (July 2018–June 2019) based on surveillance data on the provision of EC in women of reproductive age (age range of 16–55 years) in community pharmacies in Catalonia, Spain, that were a part of the sentinel pharmacy network.

### 2.1. Sampling Frame

The Catalan sentinel pharmacy network, composed of 60 community pharmacies scattered throughout the region, was constituted proportionally to a stratification of the population of Catalonia based on criteria of representativeness, ensuring a coverage of 2.5% of the Catalan population. For the representativeness analysis, we considered the basic health areas (BHAs) of each healthcare sector of the region, establishing the proportion of urban, semi-urban, and mountain pharmacies; and considering geographical, socio-economic, demographic, epidemiological, cultural, and homogeneous communication channels.

### 2.2. Data Collection

Data were obtained using an 18-item electronic form that the users answered at the point of EC provision in the sentinel pharmacies. All cases were registered by 122 pharmacists from the 60 community pharmacies. The following variables were recorded:

Code and name of sentinel pharmacy;Date of EC dispensation;National code and name of medicinal product;User sex (male, female);Age of the user;Who the medication is for (choice between personal use, partner, friend, daughter, and other family member);Postal code of the population of residence: to determine if the person searches for the medication in a nearby pharmacy;Time from unprotected sex (hours);Time since the last menstrual period (weeks);Contraceptive method normally used;First EC dispensation or not;If it is not the first dispensation, indicate the time elapsed since the last dispensation (less than 6 months, between 6 months and 1 year, more than 1 year);If it is not the first dispensation, indicate which medication was previously taken;If it is not the first dispensation, indicate if the patient has previously had any adverse reactions;Suspected adverse reactions reported to the pharmacovigilance center (yes or no);Description of pharmaceutical action performed;Availability of an EC kit consisting of a condom and additional informative material (yes or no);Observations (free text field to indicate any relevant aspect during the EC dispensation);

The data collection tool was validated through a pilot test carried out in 2016 in 21 community pharmacies in Barcelona, Catalonia, Spain [11].

All confidential information collected was recorded in the Applications Portal of the Ministry of Health, accessible by username and password through the Drugs and Pharmacy Channel website.

The participation rate was 99.6% (941 of 945 users). Users were informed and signed the informed consent to participate in the study.

### 2.3. Data Analysis

Subject characteristics that were categorical variables were summarized as counts and percentages. Continuous variables were summarized as means with standard deviations. The results related according to the type of pharmacy selected were grouped into 3 categories: urban, semi-urban, and mountain. For the statistical analysis, the χ^2^ test was used for the study of the categorical variables and the Student *t*-test to compare continuous variables. A *p*-value < 0.05 was considered statistically significant. All analyses were conducted with SPSSS software, version 18 (SPSS Inc., Chicago, IL, USA).

## 3. Results

### 3.1. Incidence Data

During the study, 941 notifications of dispensation of EC in Catalonia were received (Figure 1). The months that correspond to holiday periods experienced a higher incidence in the number of dispensations; the highest were in the month of March 2019, which coincides with Easter, with an incidence of 244 per 100,000 women of reproductive age, and the summer months of August and September 2018, with an incidence of 233 per 100,000 women of reproductive age (Figure 1).

Similarly, urban areas reported a higher incidence in the number of dispensations with respect to semi-urban and mountain areas (*p* < 0.001 and *p* < 0.05, respectively) (Figure 2).

With respect to the type of medication, LNG was dispensed in 78.9% of cases and UPA in only 21.1%. Of all the notifications for LNG and UPA, only 12.8% and 18.6% were dispensed with an EC kit, respectively.

### 3.2. User Profile

Of the medication, 80% was dispensed to women (753 cases) and 20% was collected by men (188 cases). Almost all women (95.8%) stated that the medication was for their own use; in the case of men, 92.5% said that they collected the medication for their partner and 6.4% for a female friend. The mean age of the women was 26.9 ± 7.8 years (median of 26 years), while that of the men was 27.1 ± 9.0 years (median of 25 years), with no significant differences observed.

In relation to the patients’ behavior, 55.8% of users said it was the first time that they had taken the medication (first-time user), while 44.2% stated that they had taken it on other occasions (repeat user). An increase was observed in the number of women who requested the medication when they were repeat users compared to first-time users (85.8% vs. 75.4%, *p* < 0.001). The mean age of first-time and repeat users showed no significant differences (26.7 ± 8.4 years and 27.2 ± 7.5 years, respectively, Table 1).

Almost one-quarter of repeat users (24.3%) confirmed that they used EC less than six months previously, 23.1% used it more than six months but less than one year previously, 50.7% more than one year previously, and only 1.9% did not know this information.

Of the repeat users, 28.6% did not use any regular method of contraception compared to 24.6% of first-time users, with no significant differences observed. However, the percentage of users who used condoms was lower in the repeat users compared with first-time users (56.7% vs. 64.4%, *p* < 0.05, Table 1).

Notably, an analysis subgroup corresponded to 7.7% of cases in which the EC was dispensed to minors. In this subgroup, the medication was collected by girls in 69.4% of cases, significantly lower (*p* < 0.05) than the value for the group of repeat (85.8%) and first-time users (75.4%); mean age was 16.1 ± 1.1 years (median age of 16.5 years). Of the total number of cases, 26.4% of young people said they were repeat users; specifically, 42.1% said they had used the medication within the last six months. In general, in 31.6% of cases, young people said they did not regularly use any method of contraception, while 68.4% used condoms.

### 3.3. Behavior of the Users

In 87.2% of cases, the users had collected the medication within 24 h of unprotected sex; in 9.4%, in the following 25 to 48 h; and in 3.3%, after 48 h. In this regard, no statistically significant differences were observed between first-time and repeat users.

With respect to users’ regular contraceptive method, the condom was the most widely used method (60.1%). However, 25.7% of users stated that they did not use any barrier contraceptive method.

Nevertheless, the joint analysis of the contraceptive methods used and risk behaviors was particularly interesting. We observed that 41.7% of the subgroup of first-time users who requested the medication between 25 and 48 h did not use any barrier method, increasing to 50% in the subgroup of first-time users who requested the medication more than 48 h after risky sexual intercourse. This was significantly higher than the proportion of first-time users who obtained the medication within 24 h of unprotected sex and who used natural methods (*p* < 0.01). Similarly, the use of natural methods in repeat users was 53.8% in the subgroup who requested the medication after 48 h, significantly higher than in users who obtained the medication within the first 24 h (*p* < 0.05) (Figure 3a).

In the case of condoms, 50% and 38.9% of first-time users who collected the EC in the 25–48 h interval and after 48 h, respectively, used this contraceptive method. These percentages are significantly lower than the percentage of women who obtained the medication within the first 24 h (Figure 3b).

Likewise, when users were asked about their last menstrual period, 50% of first-time users and 30.8% of repeat users who took more than 48 h to seek EC did not remember the last time they had their period. Moreover, in 8.1% of cases, the women requested the EC more than 4 weeks after their last menstrual period.

Eighty-two percent of users said that they had obtained the EC at the pharmacy in their neighborhood (Table 2).

Only 18% of cases obtained the EC in areas outside their neighborhood. No age differences were observed in the groups studied. When data from repeat users were analyzed, no differences were observed between users who obtained the medication outside their area of residence compared with the same area of residence (46.1% vs. 43.7%), while a higher percentage of women collected the medication compared to men, increasing from 80% to 86.4% in the same area of residence and to 84.4% in outside areas (Table 2).

### 3.4. Pharmacovigilance and Pharmaceutical Care

In relation to pharmacovigilance of the EC, 25 suspected adverse reactions were detected (2.7%), the most common being menstrual delay, dizziness, and general malaise.

In approximately half of dispensations (53.9%), the community pharmacist performed additional actions by providing the users with personalized information. The information provided included dispensing pharmacotherapeutic advice and guidance on STDs, the rational use of the medication, possible interactions with other medications, the most common side effects, problems related with the medication, and the use of alternative contraceptive methods. Furthermore, in complex cases, the pharmacist referred the patient to a gynecologist after providing advice on preventive measures and health education.

## 4. Discussion

Numerous studies have been conducted in recent years to determine the profile of women seeking EC, factors associated with self-reported use, and sexual behavior of users. The vast majority of these were performed using data obtained in primary care centers, hospital or primary care emergency services, and family planning clinics. In contrast, little data were obtained in community pharmacies through over-the-counter dispensing [11,12,13].

Contrasting the other studies, our study is one of the first carried out in the setting of a community pharmacy. Given the proximity and importance of the pharmacist as a first-line agent, we obtained a sufficiently representative sample that allowed us to characterize the population that uses EC from the care dynamic, improving knowledge of the use of this medication.

The results for the incidence of use of EC revealed that usage of this medication was higher during holiday periods: it was highest in March 2019, which coincides with Easter holidays, with an incidence of 244 per 100,000 women of reproductive age, and the summer months of August and September 2018, with an incidence of 233 per 100,000 women of reproductive age. Other national studies showed similar findings, reporting that the months in which EC was most requested were August, September, and December during the period from June 2002 to June 2004 [14].

Relevant incidence results were also obtained that show the behavioral pattern of use of EC in Catalonia, a region of Spain situated in the south of Europe. Notably, the incidence data per BHA obtained throughout the study year showed that the incidence of the use of EC is greater in urban versus rural areas. This contrasts the results obtained in a study conducted by the Catalan Ministry of Health during the period 2004 to 2007, in which a comparative analysis was performed between two geographical areas, concluding that, in the province of Lleida (rural area), the use of EC was higher than in the Barcelona metropolitan area (urban area) [15].

The results reveal the repeated use of this contraceptive method, since in 44.2% of cases users, said that it was not the first time they took this medication, which is consistent with other national and international studies [14,15,16].

A systematic review of the profile of EC users based on a sample of 14 articles published in Spain between 1999 and 2008 found that 9–60% of women used this medication on more than one occasion [17]. Repeat use of this medication is linked to risky sexual behavior among EC users [18,19]. In this study, the results show a higher percentage of users who do not use any regular contraceptive method and the use of condoms is lower with respect to users who take EC for the first time. Although previous studies sought to demonstrate this, the findings are inconclusive.

With respect to the age profile of the users, although we did not detect age differences between first-time and repeat users in this study, a higher overall mean age was observed in women who took EC (26.9 ± 7.8 years) in relation to data from most national and international studies, where it was 20–24 years [11,14,20,21]. However, 7.7% of EC dispensations were provided to minors, similar to that found in other studies [17]. These latter data are accompanied by a high percentage of cases that did not use any regular contraceptive method.

We observed that, in most cases, women used this medication within 24 h of having unprotected sex (87.2%), which is comparable to that obtained in other studies [14,17]. We also observed that repeat use of this medication was accompanied by more risky sexual behavior in users who take EC more than 48 h after unprotected sex compared with those who obtain it during the first few hours, including a lower percentage of users who use condoms, and a higher percentage who use no contraceptive method.

With respect to the contraceptive methods used, all studies agree that the contraceptive method most commonly used among EC users is the condom (60%), similar to the percentage in this study [14,17,20,21]. However, the percentage of users who use no contraceptive method in their sexual relationships (25.7%) is linked to a profile of users who use EC repeatedly, together with a longer time between having unprotected sex and seeking the medication, as described in other articles [14,22].

The results show a small but significant percentage of users to whom the EC was dispensed more than 4 weeks after their LMP, an aspect that requires attention and corrective measures since, according to product information sheets and therapeutic guidelines, its use after the ovulation period is not effective [1,2,3,4].

We demonstrated that there is easy access to EC, as in most cases (according to Table 2), users obtained the medication at the pharmacy in their area of residence. This also allows us to assume that the users are non-judgmental and show greater confidence with regard to healthcare professionals. Nevertheless, we found that in almost half of cases, the users who collected the medication at pharmacies outside their neighborhood used this medication repeatedly.

In addition, the Catalan sentinel pharmacy network enabled safety aspects in matters of pharmacovigilance to be reinforced by identifying suspected drug-related adverse events (DRAEs) for medicinal products marketed as EC. Fewer than 10% suspected DRAEs were reported, which is consistent with previous studies within the national setting [14]. These were mainly menstrual delay, dizziness, and general malaise, unlike other studies that reported a higher number of cases related to nausea, vomiting, and headache, described in other studies [7,8].

The demand for healthcare through low or very-low complexity processes remains mainly with pharmacies, as they are the first level of care. Accordingly, the community pharmacist has enormous potential, offering the possibility of monitoring the demand for pharmaceutical indication related with the preclinical stages of pathological processes, while having the responsibility of reporting the rational use of medicines and educating the population to adopt safe sexual behavior. Contrary to expectation, the study findings reveal low active participation of the pharmacist when dispensing the EC (53.9%), together with little medication accompanied by an EC kit. For these reasons, and as pharmaceutical care is essential for correct use of EC, more emphasis must be placed on the importance of the community pharmacist and more proactive work must be encouraged.

The main limitation of the study is that not all users were receptive to providing information at the time of dispensation of EC. However, since this occurrence was detected in the pilot test, pharmacists were trained with specific training workshops in assertive communication to address this issue [11].

Nevertheless, our study was able to show that the Catalan sentinel pharmacy network serves as a robust system for epidemiological surveillance, thus enabling data to be obtained on the profile and behavior of EC users in Catalonia. In this respect, a high percentage of repeat users was detected, with risky sexual behaviors, suggesting that new measures should be implemented to provide information on this method, together with educational and preventive strategies. It also helped to define some of the real health indicators of the general population.

## Figures and Tables

**Figure 1 jcm-10-02793-f001:**
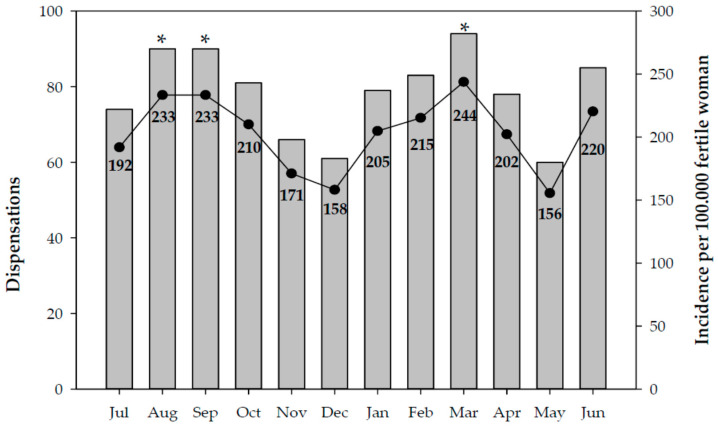
Number of EC dispensations and cumulative incidence per month. Catalonia, 2018–2019. Significant differences: * vs. the incidence in December (*p* < 0.05).

**Figure 2 jcm-10-02793-f002:**
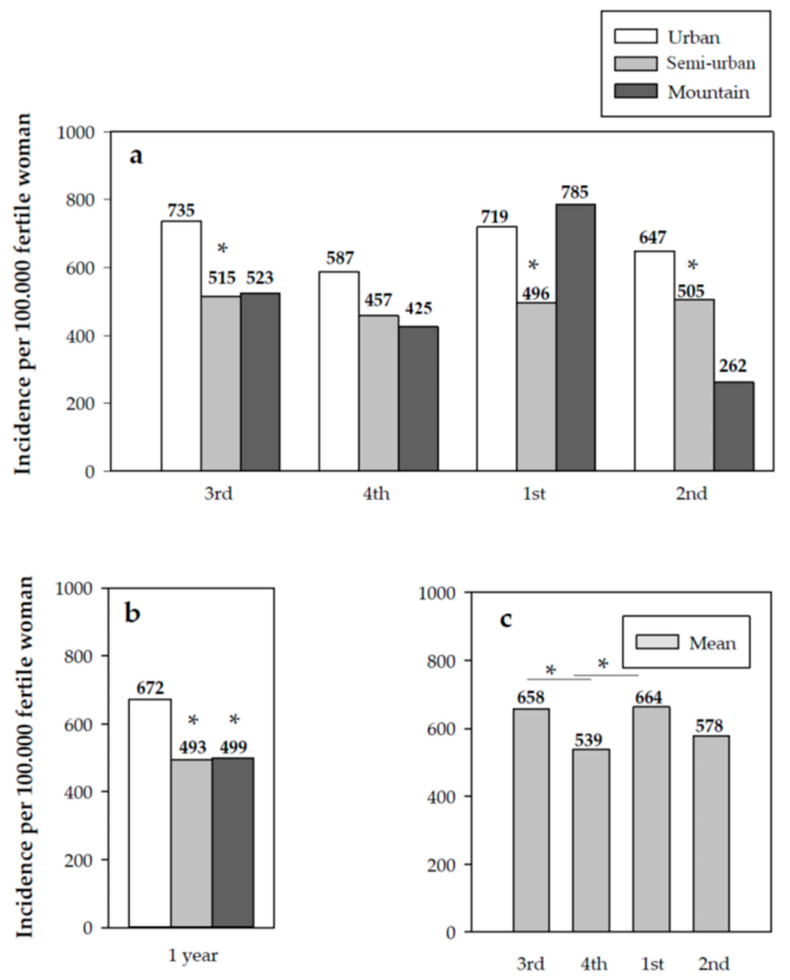
(**a**) Quarterly cumulative incidence by type of geographical area; (**b**) annual cumulative incidence by type of geographical area; (**c**) quarterly cumulative incidence in Catalonia. Significant differences: (**a**,**b**) * vs. urban area (*p* < 0.05); (**c**) * *p* < 0.05.

**Figure 3 jcm-10-02793-f003:**
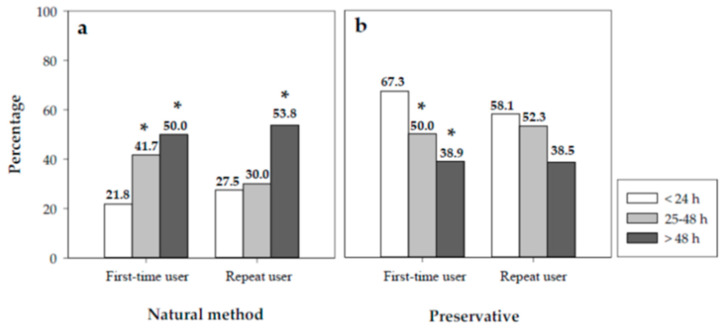
Percentages of first-time users and repeat users according to the time of collection of the medica-tion and their contraceptive behavior: (**a**) natural method, (**b**) preservative. Significant differ-ences: * vs. < 24 h (*p* < 0.05).

**Table 1 jcm-10-02793-t001:** Profile of users who used EC in Catalonia during the period 2018–2019.

Variables	Total (*n* = 941)		Medication Collected by Women (*n* = 753)		Medication Collected by Men (*n* = 188)	
	*n*	%	*n*	%	*n*	%
First-time user	525	55.8	396	75.4	129	24.6
Mean age (years)	26.7 ± 8.4		26.6 ± 8.2		27.2 ± 9.1	
Condom used	338	64.4	241	60.9	97	75.2
Natural method used	129	24.6	104	26.3	25	19.4
Repeat user	416	44.2	357	85.8	59	14.2
Mean age	27.2 ± 7.5		27.3 ± 7.3		27.0 ± 8.8	
Condom used	236	56.7	201	56.3	35	59.3
Natural method used	119	28.6	99	27.7	20	33.9

**Table 2 jcm-10-02793-t002:** Profile and behavior of users who used EC according to their area of residence.

Variables	Total		Medication Collected by Women		Medication Collected by Men	
	*n*	%	*n*	%	*n*	%
Same area of residence	759	82.0	607	80.0	152	20.0
Mean age (years)	26.9 ± 8.0		27.0 ± 7.7		26.7 ± 8.9	
Repeat user	332	43.7	287	86.4	45	13.5
Outside area of residence	167	18.0	135	80.8	32	19.2
Mean age (years)	26.9 ± 8.4		26.4 ± 8.1		29.1 ± 9.4	
Repeat user	77	46.1	65	84.4	12	15.6

## Data Availability

The datasets that support the findings of this study are available from the first author (A.M.J.) upon reasonable written request.

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
