# Peer review of "Characterization of the Use of Emergency Contraception from Sentinel Pharmacies in a Region of Southern Europe"

_jcm, 2021, doi:10.3390/jcm10132793_

Round 1

Reviewer 1 Report

Here they are my comments.

Please mention the research aim in the abstract.

Lines 102-105, BHA has been considered for analysis, as you have stated. Please in the data analysis section/results of the study, mention how it has been considered and how the findings have been presented accordingly.

Before the data collection by the sentinel pharmacist, how these data about the person who used medications, have been collected and registered?

Please describe how the data collection tool has been validated and how its reliability has been checked? How many pharmacists have collected data? How can you be sure of reliability of data collection and the presence of Bias in this process? Have you checked kappa score?

How abour ethical considerations in this research?

Add the end of the discussion, please add the limitations of this study.

Reviewer 2 Report

This is an interesting study on the use of EC in Catalonia. While several similar studies have been published on the subject, this study is characterized by a thorough collection of information about the users thanks to a network of community pharmacies.

The study is well designed, well conducted and clearly presented. All the figures are clear and useful and all the relevant references are reported.

However, before the paper is  suitable for publication, the authors should address a few points.

  1. Please, clarify what exactly is a "community pharmacy" in Catalonia; the same definition could apply to different institutions in different countries.
  2. How relevant is the network of community pharmacies in comparison to all the other pharmacies in Catalonia? (i.e.  how many community pharmacies and other pharmacies are there in Catalonia?)
  3. How many dispensations of EC were made in Catalonia during the study period considering all the pharmacies?
  4. Please comment on the fact that knowing that a study was conducted in community pharmacies might have deterred some women from using these pharmacies (did the percentage of EC prescription in community pharmacies increase, decrease or remain stable during the study period in comparison to the previous year?)
  5. If the use of EC increases during holidays, how do the authors explain that it does not increase during  Christmas season? 
  6. ULP is more effective than LNG. Is this information provided to women asking for EC in community pharmacies? If so, why is LNG more used?
  7. When the authors report about "natural method" do they refer to a fertility awareness method or to withdrawal?

Round 2

Reviewer 1 Report

The article has been improved. You provided the answers to my queries in the previous review round, but you did not incorporate them into the text. Therefore, changes should be made in the text in accordance with your answers. 

Author Response

Dear Reviewer: In the attached file we send you the modified text with the answers we sent you. We have also incorporated this version of the manuscript in place of the original manuscript. best regards
